# Enhanced IFNα Signaling Promotes Ligand-Independent Activation of ERα to Promote Aromatase Inhibitor Resistance in Breast Cancer

**DOI:** 10.3390/cancers13205130

**Published:** 2021-10-13

**Authors:** Taylor E. Escher, Prasad Dandawate, Afreen Sayed, Christy R. Hagan, Shrikant Anant, Joan Lewis-Wambi

**Affiliations:** 1Department of Cancer Biology, University of Kansas Medical Center, 3901 Rainbow Boulevard, Kansas City, KS 66160, USA; tescher@kumc.edu (T.E.E.); pdandawate@kumc.edu (P.D.); asayed@kumc.edu (A.S.); chagan@kumc.edu (C.R.H.); sanant@kumc.edu (S.A.); 2The University of Kansas Cancer Center, 3901 Rainbow Boulevard, Kansas City, KS 66160, USA; 3Department of Biochemistry and Molecular Biology, University of Kansas Medical Center, 3901 Rainbow Boulevard, Kansas City, KS 66160, USA

**Keywords:** breast cancer, aromatase inhibitor, estrogen deprivation, AI resistance, estrogen receptor, interferon alpha, interferon-stimulated genes, IFITM1, STAT1

## Abstract

**Simple Summary:**

Interferon alpha (IFNα) signaling is highly upregulated in ER+ breast cancers that become resistant to estrogen deprivation therapy. This study uncovers how enhanced (IFNα)/JAK-STAT signaling directly influences estrogen receptor (ERα) activation in the absence of estrogen. We found that inhibiting IFNα signaling downregulates expression and activation of ERα. In addition, STAT1 and ERα directly interact and regulate a key interferon-stimulated gene, IFITM1, in AI-resistant breast cancer cells. We demonstrate that crosstalk occurs between IFNα and ERα pathways, which contributes to aggression and survival of AI-resistant breast cancer, thus representing a novel mechanism of acquired AI resistance.

**Abstract:**

Aromatase inhibitors (AIs) reduce estrogen levels up to 98% as the standard practice to treat postmenopausal women with estrogen receptor-positive (ER+) breast cancer. However, approximately 30% of ER+ breast cancers develop resistance to treatment. Enhanced interferon-alpha (IFNα) signaling is upregulated in breast cancers resistant to AIs, which drives expression of a key regulator of survival, interferon-induced transmembrane protein 1 (IFITM1). However, how upregulated IFNα signaling mediates AI resistance is unknown. In this study, we utilized MCF-7:5C cells, a breast cancer cell model of AI resistance, and demonstrate that these cells exhibit enhanced IFNα signaling and ligand-independent activation of the estrogen receptor (ERα). Experiments demonstrated that STAT1, the mediator of intracellular signaling for IFNα, can interact directly with ERα. Notably, inhibition of IFNα signaling significantly reduced ERα protein expression and ER-regulated genes. In addition, loss of ERα suppressed IFITM1 expression, which was associated with cell death. Notably, chromatin immunoprecipitation experiments validated that both ERα and STAT1 associate with ERE sequences in the IFITM1 promoter. Overall, hyperactivation of IFNα signaling enhances ligand-independent activation of ERα, which promotes ER-regulated, and interferon stimulated gene expression to promote survival in AI-resistant breast cancer cells.

## 1. Introduction

Breast cancer is the most frequently diagnosed cancer among women in the United States (U.S.) and the second highest cause of death. It is estimated that in 2021, approximately 281,550 U.S. women will receive a diagnosis of invasive breast cancer and 43,000 women will die from the disease [1,2,3,4]. The most frequently diagnosed subtype of breast cancer is ER+ (estrogen receptor and/or progesterone receptor positive) accounting for approximately 70% of diagnoses [5,6]. Because estrogen promotes cancer progression, the standard treatment for ER+ breast cancer blocks ERα signaling. Aromatase inhibitors (AIs), which decrease estrogen production by 90–98%, are now the standard of care for postmenopausal ER+ breast cancer [7,8,9,10]. Despite the efficacy of AIs, approximately 30% of patients develop recurrent disease within 10 years, demonstrating resistance to AIs in tumors. To treat these patients, it is critical to understand the molecular mechanisms of acquired AI resistance.

Our lab has previously identified that the interferon alpha (IFNα)/JAK-STAT signaling pathway is a mechanism of AI resistance [11]. Interferons (IFNs) are cytokines secreted by the body to increase antiviral responses in cells, but they also influence pro-survival signaling mechanisms. Picomolar concentrations of IFNs autonomously secreted by cancer cells for an extended duration induce expression of ISGs important in cancer progression and therapy resistance [12,13]. For example, expression of ISGs including PLSCR1 and IFITM1/3 promotes tumor progression and invasion in clinical samples and cancer cell lines [14,15,16]. Mechanistically, IFNα activates JAK/STAT signaling through binding to the IFNα receptor 2 (IFNAR2). P-STAT1, P-STAT2, and interferon regulatory factor 9 (IRF9) bind to interferon-stimulated response elements (ISREs) to transcribe ISGs. Indeed, hyperactivated IFNα/JAK/STAT signaling is pro-tumorigenic and confers resistance to treatment; however, the exact molecular mechanism has yet to be elucidated [17,18].

Recently, our lab identified overexpression of multiple ISGs in AI-resistant breast cancer including interferon-induced transmembrane protein 1 (IFITM1), which is critical for regulating survival of acquired AI-resistant breast cancer cells [11,19,20]. IFITM1 contributes to protein complexes involved in cell adhesion, germ cell homing, and viral infection [21]. IFITM1 resides on plasma membranes and can interact with multiple proteins including RAB5, LAMP1, CD63, CD19, CD21 and CD81 [22,23], which can regulate EMT, cell adhesion, angiogenesis, invasion, and metastasis [24]. Other studies have validated the importance of IFITM1 and shown correlation with poor overall and recurrence-free survival in multiple tumor types. Its overexpression in gastric, esophageal, colorectal, cervical, ovarian, brain, and breast cancer promotes proliferation, migration, invasion, and metastasis [25,26,27,28,29,30,31,32,33].

Traditionally, the estrogen receptor functions as a classic steroid hormone receptor. Estrogen binds to ERα, allowing it to act as a transcription factor [34]. ERα can also signal through a ligand-independent (or estrogen-independent) mechanism. Through tethered protein–protein interactions with other transcription factors (FOXA1 SP1, AP1, Pitx1, Runx1, SF-1, SRCs, NFκB, and C/EBP) and rapid phosphorylation cascades through the serine 167 and 118 residues, ERα can regulate signaling non-canonically [34,35,36,37]. Many cell signaling proteins including GPCRs, Src, PI3K, EGFR, HER2, IGF-1, and MAPK can influence the ligand-independent activation of ERα [34,38]. This ligand-independent signaling allows breast cancers to become resistant to AIs. We hypothesize that long-term estrogen deprivation enhances IFNα signaling in AI-resistant breast cancer cells, which drives ligand-independent activation of ERα and promotes AI resistance.

In this study, we investigated whether ERα is directly regulated by IFNα signaling, representing a potential mechanism by which ER+ breast cancers develop resistance to estrogen deprivation therapy. We discovered that AI-resistant MCF-7:5C breast cancer cells constitutively express enhanced ERα (total and phosphorylated) expression as well as ER-regulated genes compared to AI-sensitive MCF-7 and T47D cells and that blockade of IFNα signaling or knockdown of STAT1/2 markedly reduces ERα expression and ER-regulated genes in these cells. Additionally, we found that ERα and STAT1 physically interact in MCF-7:5C cells through protein docking studies, proximity ligation assay, and immunoprecipitation. Notably, chromatin immunoprecipitation studies show that STAT1 and ERα associate with IFITM1 promoter to drive its expression in AI-resistant MCF-7:5C cells. Surprisingly, ligand treatment (through E_2_) inhibits IFITM1 promoter occupation and expression and induces cell death. Together, these findings demonstrate a novel crosstalk between IFNα signaling and ligand-independent activation of ERα in promoting AI resistance in breast cancer.

## 2. Materials and Methods

### 2.1. Cell Lines

The MCF-7 cell line was obtained from Dr. V. Craig Jordan (University of Texas MD Anderson Cancer Center, Houston, TX, USA) and the MCF-7:5C was cloned from parental MCF-7 cells following long-term (>12 months) culture in estrogen-free medium [39]. The T-47DA:18 cell line (hereafter referred to as T47D) was derived from T47D cells originally obtained from ATCC (Rockville, MD, USA) [40,41]. All cell lines were cultured as previously described [42] at 37 °C under 5% CO_2_.

### 2.2. Small Interfering RNA (siRNA) Transfections

T47D, MCF-7, and MCF-7:5C cells (1 × 10^5^) were transiently transfected with siRNAs for ERα (Santa Cruz Biotechnology, Dallas, TX, USA, Cat#sc-29305), STAT1 (Santa Cruz Biotechnology, Cat#sc-44123), and STAT2 (Santa Cruz Biotechnology, Cat#sc-29492) or a scrambled negative control (Santa Cruz Biotechnology, Cat#sc-37007). The ERα, STAT1, STAT2 and control siRNAs were pools of three target-specific 20 to 25 nt siRNAs as previously described [42].

### 2.3. Cell Counting for Proliferation

T47D, MCF-7, and MCF-7:5C (1 × 10^4^) cells were assayed for viability and proliferation in 24-well plates in triplicate in estrogen-free medium. After 72 h transfection, cells were counted by Trypan blue (Sigma, St. Louis, MO, USA, Cat#T8154) exclusion direct cell counts.

### 2.4. Western Blotting

Following 48 h treatment as indicated with 1 nmol E_2_ (Sigma, Cat#E8875), 48 h treatment with siER (Santa Cruz Biotechnology, Cat#SC-29305), 24 h treatment with 10 µmol IFNAR NAb (Millipore, Burlington, MA, USA, MAB1155), 48 h treatment with 1 µmol Ruxolitinib/Jakafi™ (Rux) [43] as indicated (Cayman Chemical, Ann Arbor, MI, USA, Cat#11609), cells were harvested, underwent protein assay, separated by SDS-PAGE, transferred and blocked. After primary and secondary antibody incubation, bands were detected and exposed to film, as previously described [42]. Target proteins were detected using primary antibodies: anti-p-ERα S167 (Cell Signaling, Danvers, MA, USA, Cat#64508S), anti-p-ERα S118 (Cell Signaling, Cat#2511S), anti-ERα (Cell Signaling, Cat#8644S), anti-p-STAT1 (Santa Cruz Biotechnology, Cat#SC-8394), anti-p-STAT2 (Cell Signaling, Cat#88410S), anti-STAT1 (Santa Cruz Biotechnology, Dallas, TX, USA, Cat#SC-464), anti-STAT2 (Santa Cruz Biotechnology, Cat#SC-514193), anti-IFITM1 (Santa Cruz Biotechnology, Cat#SC-374026) or anti-β-actin (Cell Signaling, Cat#3700S). Western blotting of proteins is available in Appendix A.

### 2.5. RNA Isolation and Real-Time PCR

Following 48 h treatment or transfection, cells were harvested, total RNA was isolated, cDNA was synthesized, and RT-PCR was conducted. (Primers are outlined in Appendix A [42]). Relative mRNA expression level was determined as the ratio of the signal intensity to that of PUM1 using the formula: 2^−ΔCT^. When cells were treated, fold change in gene expression was normalized to PUM1 and then compared to the untreated value for that cell line using the formula: 2^−ΔΔCT^.

### 2.6. Immunofluorescent (IF) Staining

Following fixing and permeabilization, cells were stained using primary antibodies against anti-IFITM1 (Santa Cruz Biotechnology, Cat#SC-374026) and anti-ERα (Santa Cruz Biotechnology, Cat#SC-544) followed by secondary antibodies, and mounted as previously described [42]. Images were collected on a Leica TCS SPE confocal microscope and analyzed using the Leica LAS AF Lite software (Leica Biosystems, Nussloch, Germany).

### 2.7. Proximity Ligation Assay

Fixed cells were washed with PBS, permeabilized with 0.1% Triton-X in 1× PBS for 15 min, and incubated with Duolink^®^ blocking solution. For the remainder of the processing the Duolink^®^ PLA Fluorescence (Sigma Aldrich, St. Louis, MO, USA, #DUO92001) was used per the manufacturer’s instructions. Antibodies used were ERα rabbit (Cell Signaling Technologies, #8644S,) and STAT1 mouse (1:200, Santa Cruz: sc-464).

### 2.8. Dual Luciferase Reporter Assays

For promoter assays, 0.8 μg of plasmid DNA and pRL CMV Renilla vector were used [44]. For analysis of IFITM1 promoter activity, the pGL3 plasmid with the first 750 nucleotides of the IFITM1 promoter inserted (pGL3-IFITM1 [−750/−1]) was used [44]. The pGL3-Basic-IRES was a kind gift from Joshua Mendell (Addgene, Watertown, MA, USA, Cat#64784) [45]. Analysis of ISRE and ERE promoter activity was previously described [42].

### 2.9. TUNEL Staining

After 72 h transfection, TUNEL staining was conducted using the Click-iT Plus TUNEL Assay Kit (Invitrogen, Waltham, MA, USA, Cat#C10618) following the manufacturer’s instructions. The average TUNEL intensity was quantified using the red color channel on Image J software for a minimum of three images.

### 2.10. Co-Immunoprecipitation (Co-IP)

Cell lysates were collected incubated overnight at 4 °C with 2 µg appropriate antibody or control IgG. 50:50 Protein A/G coated magnetic beads (Invitrogen, Cat#10001D/Cat#10003D) were then added for the final 1 h of incubation time. Immune complexes were washed three times with PBS, resuspended in Laemmli sample buffer (Invitrogen, Cat#NP0007), boiled for 5 min, and subjected to Western blotting analysis.

### 2.11. In Silico Docking Analysis

X-ray crystal structures of STAT1 (1YVL) and ERα (1A52) were downloaded from the protein data bank (PDB). These PDB files were prepared for docking analysis by removing ligands, water molecules and extra chains of amino acids that may interfere with the protein–protein interactions. Chain A was selected for both proteins which were further energy minimized to ensure the best folding of each protein and polar hydrogens and Kollman Charges were added using MGL tools. The final prepared protein structures were uploaded to the GRAMM-X Protein–protein Docking Web Server v.1.2.0 (http://vakser.compbio.ku.edu/resources/gramm/grammx/, accessed on 8 February 2021) to check for protein–protein interactions [46,47]. The final output file was analyzed using the PYMOL program to isolate the interacting amino acids and bond lengths [48].

### 2.12. Chromatin Immunoprecipitation (ChIP) Assay

ChIP was performed after sonication using the ChIP-IT Express Kit (Active Motif, Carlsbad, CA, USA, Cat#53008) according to the manufacturer’s instructions. Lysates were immunoprecipitated (IP) overnight (18 h) with the following antibodies: anti-STAT1 (Cell Signaling, Cat#9172S), anti-ERα (Cell Signaling, Cat#8644S) or an equal amount of rabbit IgG (Santa Cruz Biotechnology, Cat#SC-2027). Resulting DNA was analyzed using qPCR and run on a DNA gel. Data are represented as a percentage of input DNA.

## 3. Results

### 3.1. ERα and ER-Regulated Genes Are Upregulated in Aromatase Inhibitor-Resistant Breast Cancer Cells

The estrogen-independent MCF-7:5C clone was derived from MCF-7 cells after long-term estrogen deprivation [39,49]. MCF-7:5C cells maintain wild-type ERα but lose PR expression and exhibit estrogen (E_2_)-induced cell death [50,51]. To better understand ERα function in AI-resistant MCF-7:5C cells, we measured total and phosphorylated ERα expression in these cells compared to AI-sensitive MCF-7 and T47D breast cancer cells. We found that AI-resistant MCF-7:5C cells expressed markedly elevated levels of total and phosphorylated (S167 and S118) ERα compared to MCF-7 and T47D cells and that ERα was primarily localized in the nucleus in MCF-7:5C cells, thus indicating a constitutively activated state (Figure 1A,B). Next, we examined the mRNA expression levels of five ER-regulated genes, including cyclin D1 (CCND1), c-Myc, Cathepsin D (CTSD), pS2, and FOXA1. As shown in Figure 1C, all the ER-regulated genes were highly upregulated in AI-resistant MCF-7:5C cells compared to MCF-7 and T47D cells, despite these cells being grown in estrogen-free conditions (Figure 1C). Finally, we assessed the mRNA expression of several well-known ER-coregulatory proteins including SP1, SRC-1, SRC-3, CBP, p300, GATA3, and CITED1 in MCF-7, T47D, and MCF-7:5C cells (Figure 1D). Surprisingly, we found that T47D cells had the highest expression of ER-coregulatory proteins, whereas MCF-7:5C cells had the lowest level compared to MCF-7 cells (Figure 1D).

### 3.2. Loss of ERα Expression Induces Apoptosis Most Prominently in Aromatase Inhibitor-Resistant Breast Cancer Cells

To test whether loss of ERα expression significantly impacts the phenotype of MCF-7:5C, MCF-7, and T47D cells, we used siRNA to inhibit its expression. Figure 2 shows that loss of ERα markedly reduced the growth of AI-resistant MCF-7:5C cells compared to MCF-7 and T47D cells and TUNEL staining confirmed that the decrease in growth was due to apoptosis, which was most pronounced in MCF-7:5C cells (Figure 2A,B). Immunoblotting analysis verified that ERα inhibition increased PARP cleavage primarily in MCF-7:5C cells (Figure 2C). Lastly, we observed that in all three cell lines, knockdown of ERα significantly reduced the expression of ER-regulated genes; however, the effect was most pronounced in AI-resistant MCF-7:5C cells, which expressed the highest basal level of ER-regulated genes (Figure 2D, left panel) compared to MCF-7 (Figure 2D, right panel) and T47D cells (Figure 2D, bottom panel).

### 3.3. Enhanced IFNα Signaling Affects ERα and ER-Regulated Gene Expression in AI-Resistant Breast Cancer Cells

Previously, we demonstrated the enhanced IFNα signaling in AI-resistant MCF-7:5C cells [11,26]; hence, we investigated the consequence of enhanced IFNα signaling on ERα function and its transcriptional activation. We first measured the expression of multiple interferon-stimulated genes (ISGs) including IFNα, IFNβ, IFIT1, IRF9, OAS1, STAT1, STAT2, PLSCR1, and IFITM1 and found that they were markedly elevated in AI-resistant MCF-7:5C cells but not expressed in MCF-7 cells (Figure 3A). Next, we determined whether enhanced IFNα signaling alters ERα expression and function by blocking the IFNα signaling pathway using an IFNAR neutralizing antibody (IFNAR NAb) and a JAK1 inhibitor, Ruxolitinib (Rux) (Figure 3B–D). Blockade of IFNα signaling significantly reduced total ERα, p-ERα S167 levels, p-STAT1/p-STAT2, and IFITM1 expression (Figure 3B) in AI-resistant MCF-7:5C cells. The expression of our selected ER-regulated genes (CCND1, pS2, CTSD, FOXA1, and c-Myc) was markedly reduced in these cells with no significant effect in MCF-7 cells (Figure 3C). Finally, blockade of IFNα signaling also reduced ERE (estrogen response element) activity in AI-resistant MCF-7:5C cells, as demonstrated by the luciferase assay shown in Figure 3D. Confirmation of the inhibitors effect on STAT1 and STAT2 at the mRNA level in all cell lines is shown in Appendix A along with two downstream targets, IFITM1 and IRF9. The same experiments were performed in T47D cells and show little impact on ERα signaling (Appendix A).

### 3.4. STAT1 and STAT2 Expression Affect ERα and ER-Regulated Gene Expression in AI-Resistant Breast Cancer Cells

Since the JAK/STAT inhibitor Rux dramatically reduced ERα expression in AI-resistant MCF-7:5C cells, we next determined whether inhibition of STAT1 and STAT2 directly impacts ERα expression and function in these cells. We utilized siRNAs to target STAT1 and STAT2 expression in MCF-7:5C and MCF-7 cells. Inhibition of STAT1 and STAT2 reduced total ERα and p-ERα levels in MCF-7:5C cells but not MCF-7 cells (Figure 4A). Notably, loss of STAT1 significantly reduced ERE activity in MCF-7:5C cells (Figure 4B) while loss of STAT1 and STAT2 dramatically reduced the expression of ER-regulated genes in MCF-7:5C cells (Figure 4C, right panel) with no effect in parental MCF-7 cells (Figure 4C, left panel). Confirmation of STAT1 and STAT2 knockdown is shown in Figure 4D along with two downstream targets IFITM1 and IRF9. Similar experiments were performed in T47D cells and although ERα protein level was reduced, ERE luciferase activity and ER-regulated genes were not impacted (Appendix A).

### 3.5. STAT1 Interacts with ERα through In Silico and In Vitro Analysis in AI-Resistant Breast Cancer Cells

Our data validated that STAT1 can alter ERα levels, thus we sought to determine whether these proteins physically interact through in silico analysis using the GRAMM-X Protein–protein Docking Web Server v.1.2.0. We found a potential binding site between STAT1 and ERα (Figure 5A). The STAT1 amino acids (denoted in pink) D292, S307, S315, and T489 interact with L544, G339, P337, L416, N413, and M437 of ERα (denoted in green), respectively. Binding occurs within 2.2 to 3.5 angstroms, indicating strong bonds and a relatively stable complex between the STAT1 DNA binding domain and ERα AF2 domain. This suggests that STAT1 may be interacting with ERα in lieu of ligand to cause its activation. In vitro, we validated the interaction between STAT1 and ERα through two methods. First, we performed immunoprecipitation of ERα and used Western blotting to detect protein interactions. Only STAT1, not STAT2 or IRF9, interact with ERα in the MCF-7:5C cells (Figure 5B). Finally, we utilized proximity ligation assay for STAT1 and ERα and saw predicted interactions solely in the AI-resistant MCF-7:5C cells (Figure 5C). Overall, this finding indicates a novel interaction between STAT1 and ERα in the AI-resistant MCF-7:5C cell line.

### 3.6. Inhibition of ERα Directly Affects IFITM1, a Downstream Target of IFNα Signaling

We previously reported that interferon-induced transmembrane protein 1 (IFITM1) is a key regulator of growth and survival in AI-resistant MCF-7:5C cells [11] and its expression correlates with ERα expression. In this experiment, we utilized siRNA knockdown to examine the effect of ERα loss on IFITM1 expression. We found that inhibiting ERα markedly reduced IFITM1 expression at the protein (Figure 6A) and the mRNA level (Figure 6B) and it decreased IFITM1 reporter activity and interferon-stimulated response element (ISRE) luciferase activity in MCF-7:5C cells but not MCF-7 or T47D cells (Figure 6C and Appendix A). Notably, loss of ERα also significantly reduced p-STAT1 but not p-STAT2 levels in MCF-7:5C cells, thus confirming an important role for ERα/STAT1 crosstalk in regulating IFITM1.

### 3.7. ERα and STAT1 Regulate IFITM1 through Binding to ERE and ISRE Elements in the Promoter

To investigate whether ERa and STAT1 directly bind to the IFITM1 promoter, we first determined whether there were any ERE binding sites in the IFITM1 promoter. Utilizing the UCSC Genome Browser, we searched previously uploaded chromatin immunoprecipitation data from Tamoxifen-resistant cells for ERE-like sequences. We found multiple ERE-like sequences (AGGTCACCCTGACCT) within the IFITM1 promoter at 30 kB upstream and 45, 85, and 100 kB downstream of the start site (Figure 7A). Next, we performed chromatin immunoprecipitation in all three of our cell lines with primers selective for each of the EREs and the well-known ISRE in the IFITM1 promoter (Figure 7B and Appendix A). We found that STAT1 and ERα were recruited to the ERE of IFITM1 in AI-resistant cells and to the ISRE. In MCF-7:5C cells the maximum ERα binding occurred at the 100 kB ERE (3.0×) with the lowest binding at 45 kB ERE (1.9×) when compared to the IgG. The maximum STAT1 binding was observed at the 100 kB ERE site (2.7×) and the lowest binding at the 45 kB ERE (1.7×) compared to the IgG. Overall, these data suggest that both STAT1 and ERα bind to the ERE sites within the IFITM1 promoter as well as the ISRE element. The strongest binding of both proteins occurs at the 100 kB ERE site. The MCF-7 and T47D cells (Appendix A) did not have significant binding of STAT1 or ERα to any of these sites within the IFITM1 promoter.

### 3.8. E_2_ Treatment Inhibits IFITM1 Expression and Blocks ERα and STAT1 Recruitment to the IFITM1 Promoter

Based on the observation that ERα was transcriptionally active in our AI-resistant MCF-7:5C cells and that ERα/STAT1 interaction regulates IFITM1 expression in these cells, we assessed the impact of E_2_ on ERα function and ER-mediated regulation of IFITM1 expression. Western blot and RT-PCR analyses revealed that E_2_ treatment completely reduced IFITM1 protein (Figure 8A) and IFITM1 mRNA (Figure 8B) expression in AI-resistant MCF-7:5C cells. Additionally, ChIP assays revealed that there was significant ligand-independent recruitment of ERα and STAT1 to the IFITM1 promoter at the location of the ERE sites in MCF-7:5C cells; however, in the presence of E_2_, the binding of both STAT1 and ERα to the IFITM1 promoter was markedly reduced (Figure 8C). Taken together, these data indicate that in AI-resistant MCF-7:5C cells, E_2_ acts in a paradoxical manner to suppress ERα function and thus block its potential crosstalk with the IFNα signaling pathway.

## 4. Discussion

Estrogen deprivation through AIs is the first-line treatment for ER+ breast cancer patients; however, resistance develops in approximately 30% of patients. Understanding the mechanism by which AI resistance occurs is of great clinical importance. We sought to investigate the mechanisms of AI resistance in our MCF-7:5C cell line. Previous findings from our lab demonstrate enhanced IFNα signaling in AI-resistant cells, which is key in mediating cell survival. In this study, we show for the first time that IFNα signaling is enhanced in AI-resistant MCF-7:5C cells and promotes ligand-independent regulation of ERα primarily through STAT1. This novel interaction may explain why these AI-resistant cells retain expression of ERα and continue to grow in estrogen-free conditions. Pharmacologically targeting the interaction between IFNα signaling and ERα may improve survival outcomes of patients with AI-resistant breast cancer.

The AI-resistant MCF-7:5C cells show enhanced expression of ERα compared to the MCF-7 and T47D AI-sensitive cells (Figure 1A). In addition, they have elevated levels of p-ERα at both the S167 and S118 residues and primarily nuclear localization of ERα, indicating that ERα is in an activated state (Figure 1A,B). Enhanced phosphorylation of ERα through coactivators can drive AI-resistant survival of breast cancer cells [52]. When ERα is activated (via ligand binding or phosphorylation), it begins transcription of its target genes. We saw enhanced expression of multiple ER-regulated genes (CCND1, pS2, CTSD, FOXA1, and c-Myc) in the AI-resistant cells (Figure 1C). Surprisingly, in the AI-resistant MCF-7:5C cells, the ERα coactivators (SP1, SRC1, SRC3, CBP, P300, GATA3, and CITED1) had much lower basal activation (Figure 1). This could indicate that in AI-resistant cells, JAK/STAT signaling is acting as a coactivator in lieu of these other proteins. It is paramount to reiterate that the expression and localization of ERα occurs in the absence of estrogen in the AI-resistant MCF-7:5C cells. In comparison, the levels of ERα in MCF-7 and T47D cells are shown under estrogen containing conditions (Figure 1). This suggests that the AI-resistant MCF-7:5C cells have a ligand-independent mechanism of activating ERα via phosphorylation that turns on ER-regulated genes. We hypothesize that this is through the enhanced IFNα signaling previously demonstrated in this cell line and that IFNα signaling is compensating for the loss of estrogen and continuing to activate ERα, which promotes AI-resistant cell survival.

Preclinical and clinical studies suggest an interaction between the IFNs and estrogen signaling pathways that leads to progression of breast cancer [53,54,55,56]. These studies highlight IRF-1, a target gene of JAK-STAT signaling, in acquired anti-estrogen and tamoxifen resistance [57,58,59,60]. This would offer a possible explanation as to why many breast tumors retain ERα expression but become resistant to treatment targeting ERα signaling. ERα is known to induce transcription of Jak2 in MCF-7 cells and the STAT proteins in endothelial cells and to promote IRF7-dependent expression of IFNα in plasmacytoid dendritic cells [35,61,62]. In addition, STAT1/3/5 has been shown to influence ERα in cancer and other diseases and the ISG, IFI27, can directly downregulate ERα expression [63,64,65]. A recent study from Hou et al. identified the involvement of STAT1 in facilitating ERα transcription in the Tamoxifen-resistant MCF-7: LCC2 cell line and this cell line showed overexpression of multiple ISGs, including IFITM1 [66]. This supports our hypothesis that IFNα can promote breast cancer progression through enhanced ERα signaling. Notably, our PLA and co-immunoprecipitation experiments indicate a direct interaction between ERα and STAT1 in AI-resistant MCF-7:5C cells, which was not observed in AI-sensitive MCF-7 and T47D cells (Figure 5). We have previously investigated the interaction between MUC1 and STAT1 in AI-resistant cells [42]. MUC1 is a well-known regulator of ERα and may facilitate the binding of STAT1 in activating ligand-independent signaling of ERα; however, this requires further investigation.

We found that blocking IFNα and JAK/STAT signaling through genetic and pharmacological mechanisms inhibits ERα expression (both total and S167 levels) and ER-regulated genes in the MCF-7:5C cells (Figure 3 and Figure 4). This indicates that JAK/STAT activation contributes to ERα phosphorylation. S118 and S167 reside in the AF-1 domain of ERα and upon phosphorylation contribute to ligand-independent activation and transcription of ER-regulated genes, including recruitment of co-activators [67]. Crosstalk with other signaling pathways [52,68,69] and mutations within the ligand binding domain of ERα can also cause ligand-independent activation and lead to therapy resistance [70,71,72]. We should note, however, that AI-resistant MCF-7:5C cells express only the wild-type ERα and not any of the previously published mutant ERα variants. There is also evidence that progesterone receptor (PR), another steroid receptor, suppresses expression of ISGs [73,74]. However, with long-term estrogen deprivation, MCF-7:5C cells lose PR expression. We hypothesize that without PR, this suppression is blocked, enhancing IFNα signaling and ISG expression possibly contributing to ligand-independent activation of ERα.

ERα plays a key role in modulating many signaling pathways. Here, we show that ERα directly regulates the IFNα signaling pathway and its downstream target IFITM1. IFITM1 has proven key in our AI-resistant cells in promoting their growth and survival [11,26]. In this study, downregulation of ERα caused a reduction in IFITM1 mRNA and protein expression and promoter activity. We went on to probe previous chromatin immunoprecipitation data using the UCSC Genome Browser, for potential ERα binding sites near the *IFITM1* promoter. In canonical signaling, ERα binds to estrogen response elements (EREs) or ERE half-sites of promoters of estrogen-responsive genes promoting cell cycle progression and growth [34,75]. However, one-third of genes regulated by ERα lack ERE sequences and ERα non-canonically regulates signaling by binding to GC-rich promoter sequences and through tethered protein–protein interactions [34,35,36,37]. Our ChIP data confirmed that ERα directly binds to multiple ERE sites within the *IFITM1* promoter and can bind in conjunction with STAT1 at these sites and the ISRE element in the AI-resistant MCF-7:5C cells (Figure 7). Hence, this finding suggests that ERα may regulate ISGs to promote cell survival and it indicates a potential crosstalk between ERα and the JAK/STAT signaling pathway in driving this process.

AI-resistant MCF-7:5C cells grow robustly in the absence of estradiol (E_2_); however, in the presence of E_2_, these cells undergo in vitro and in vivo cell death [40,41,42,51,76,77]. The cytotoxic effect of estrogen on AI-resistant cells have been verified through multiple pre-clinical studies [11,76] and low-dose estrogen is being clinically investigated in patients with resistant breast cancer [78,79,80,81]. Consistent with this paradoxical action of estrogen, we found that E_2_ treatment suppresses IFNα signaling and IFITM1 expression and it diminishes some ERα-mediated actions in AI-resistant MCF-7:5C cells. Notably, estrogen treatment also reduces activation of ERα and blocks ERα and STAT1 binding to the *IFITM1* promoter at both the ISRE and the ERE sites (Figure 8). The ability of E_2_ to suppress IFNα/JAK-STAT signaling in AI-resistant MCF-7:5C cells highlights the importance of this pathway in promoting survival in these cells and it supports previous data demonstrating the paradoxical action of E_2_ in these cells.

The current study suggests that identifying tumors with high expression of IFNα signaling, through a panel of ISGs, before second-line therapy could further improve treatment outcomes for AI-resistant breast cancer patients. These AI-resistant tumors could then be treated with inhibitors of the IFNα signaling pathway (via an IFNAR NAb or JAK/STAT inhibitor), thus blocking the IFNα signaling pathway. In addition, combination therapy using exogeneous estrogen would further suppress the IFNα signaling pathway and promote estrogen-induced apoptosis. Individually, these treatments have been studied clinically [43,78,82]; however, the combination of low-dose estrogen and Ruxolitinib (Jakafi™) has yet to be investigated in AI-resistant breast cancer. Based on our data, we hypothesize that combination treatment may be most effective to maintain tumor regression in a subset of AI-resistant patients. One limitation of these studies is that the mechanisms of resistance we investigated are restricted to MCF-7-derived cells. Other models of AI resistance (such as T47D cells) lose expression of ERα. To study the effects of IFNα signaling on ligand-independent activation of ERα, additional ER+ AI-resistant models need to be developed and investigated. Our studies are also limited as they were conducted in breast cancer cells free from the effects of the tumor microenvironment, which has significant effects on tumor progression and therapy resistance [52,69,83]. This limits the ability to translate these findings into the clinical setting. However, our previous research has demonstrated that estrogen and Rux treatment in vivo effectively reduces tumor progression [42]. It is critical for future studies to understand how crosstalk between E_2_, ERα, and IFNα signaling are mediated by the cells within the tumor microenvironment.

## 5. Conclusions

Overall, our data show that a unique phenotype exists in our AI-resistant MCF-7:5C cells. Upon long-term estrogen deprivation, hyperactivation of the IFNα signaling pathway leads to not only activated JAK/STAT signaling and overexpression of ISGs, but also ligand-independent activation of ERα. This may be through activation of ERα via phosphorylation of S167 and through co-activation of ERα by STAT1. This continues to cause increased proliferation of the AI-resistant cells through enhanced expression of many pro-survival proteins that are regulated by ERα including IFITM1 (Figure 9). This novel interaction shows that AI-resistant cells can be targeted through multiple mechanisms. Estrogen, the JAK/STAT inhibitor, Ruxolitinib, or inhibitors of other ERα and JAK/STAT coactivators could effectively treat AI-resistant patients or IFITM1-expressing breast cancer in combination [42]. Future studies are needed to determine the exact role that IFNα signaling has on ERα signaling in AI-resistant cells and how the tumor microenvironment impacts this crosstalk.

## Figures and Tables

**Figure 1 cancers-13-05130-f001:**
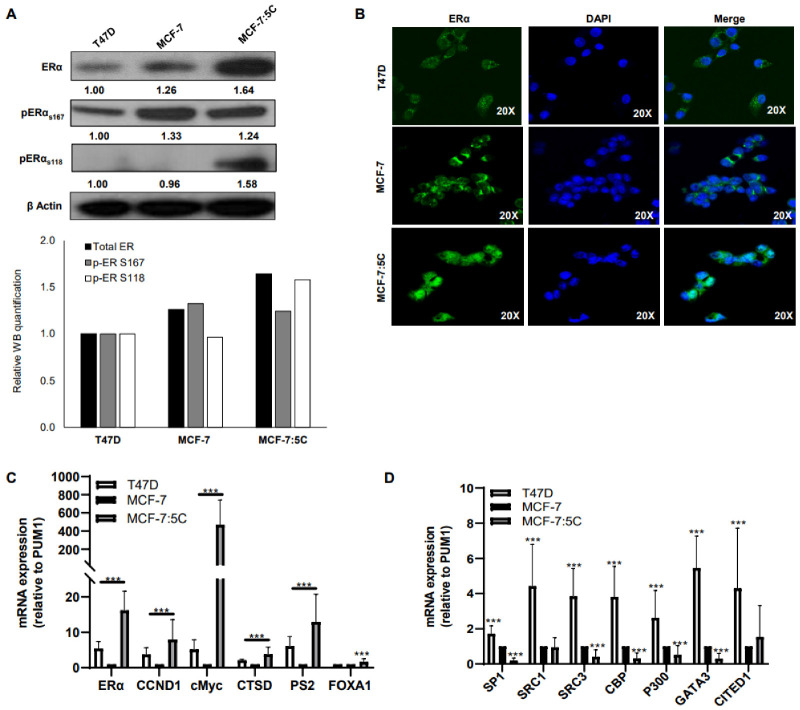
ERα and ER-regulated genes are overexpressed in aromatase inhibitor-resistant breast cancer cells. Cell lysates were subjected to (**A**) Western blot for ERα, p-ERα S167, and p-ERα S118 protein expression. Image J software was used to quantify levels of each protein relative to T47D. Quantification is shown below blot. (**B**) Immunofluorescent imaging. (**C**) RT-PCR for ERα, and the ER-regulated genes, CCND1, C-Myc, CTSD, pS2 and FOXA1 mRNA. (**D**) Transcript levels of ERα coactivators of ERα, SP1, SRC1, SRC3, CBP, P300, GATA3 and CITED1, were determined by RT-PCR. Data represent three independent experiments run in triplicate. *** *p* < 0.001.

**Figure 2 cancers-13-05130-f002:**
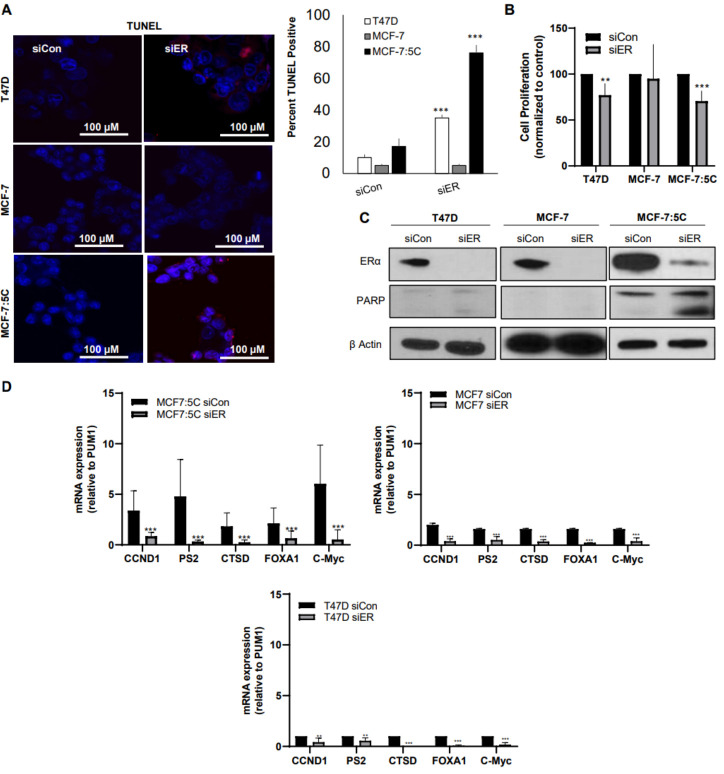
Loss of ERα expression induces apoptosis most prominently in aromatase inhibitor-resistant breast cancer cells. T47D, MCF-7, and MCF-7:5C cells were transiently transfected with siCon or siER and (**A**) measured for apoptosis by TUNEL staining, which was quantified with Image J Software (right panel); (**B**) assessed for cell proliferation c using Trypan blue exclusion 72 h after transfection; (**C**) immunoblotted for ERα and PARP expression; (**D**) analyzed by RT-PCR for mRNA expression of ER-regulated genes (CCND1, pS2, CTSD, FOXA1 and C-myc). Data represent three independent experiments run in triplicate. ** *p* < 0.05 and *** *p* < 0.01.

**Figure 3 cancers-13-05130-f003:**
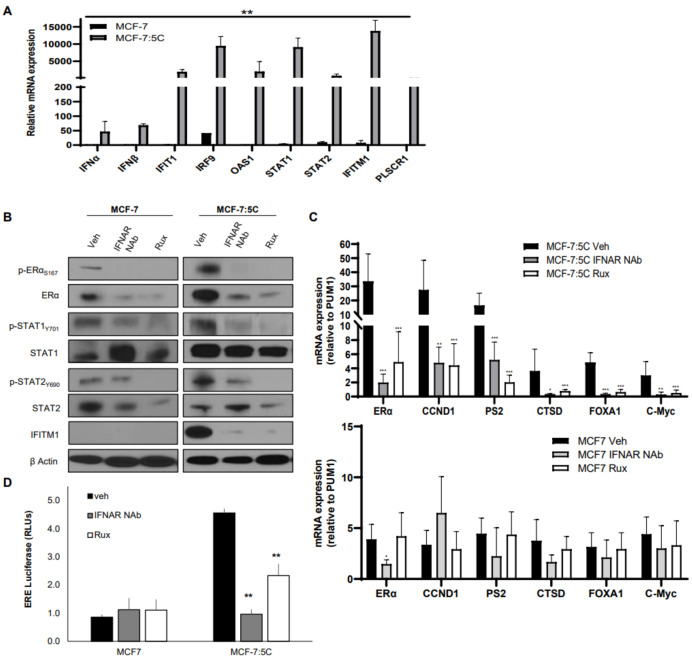
Enhanced IFNα signaling affects ERα and ER-regulated gene expression in AI-resistant breast cancer cells. (**A**) MCF-7 and MCF-7:5C cells were analyzed by RT-PCR for interferon-stimulated gene (ISG) expression. (**B**,**C**) MCF-7 and MCF-7:5C cells were treated for 48 h with IFNAR NAb or Rux (as indicated) and immunoblotted for the indicated proteins or analyzed by RT-PCR. (**D**) MCF-7 and MCF-7:5C cells were transfected with the ERE luciferase construct and then treated with IFNAR NAb or Rux. Luciferase activity was then read. * *p* < 0.1, ** *p* < 0.05 and *** *p* < 0.01.

**Figure 4 cancers-13-05130-f004:**
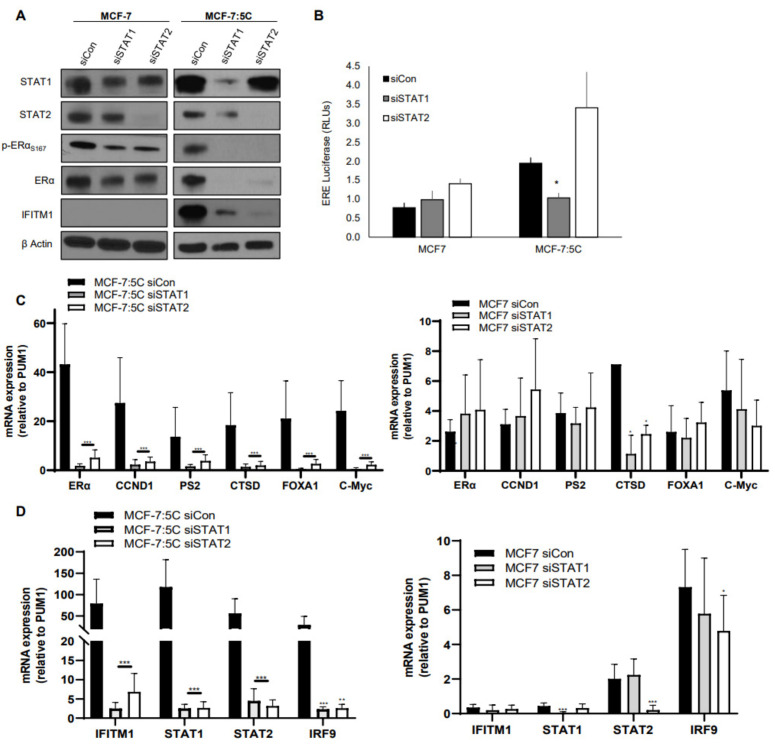
STAT1 and STAT2 expression affect ERα and ER-regulated gene expression in AI-resistant breast cancer cells. (**A**) MCF-7 and MCF-7:5C cells were transiently transfected for 48 h with siRNA against STAT1 or STAT2 and immunoblotted for the proteins indicated. (**B**) MCF-7 and MCF-7:5C cells were transiently transfected for the ERE reporter construct and siRNA against STAT1 or STAT2 for 48 h. Luciferase activity was then read. (**C**,**D**) MCF-7 and MCF-7:5C cells were transfected with siCon, siSTAT1, or siSTAT2 (as indicated) and analyzed by RT-PCR. * *p* < 0.05, ** *p* < 0.01 and *** *p* < 0.001.

**Figure 5 cancers-13-05130-f005:**
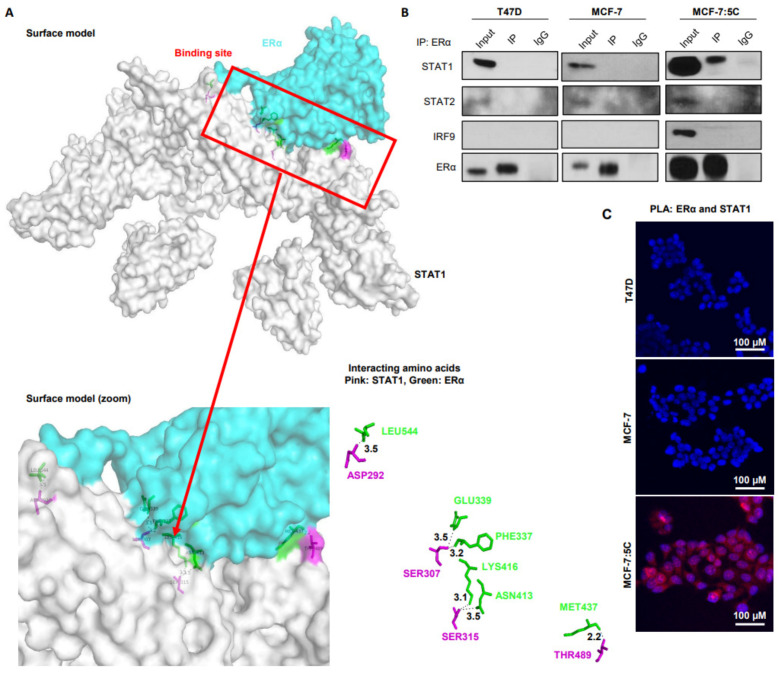
STAT1 interacts with ERα through in silico and in vitro analysis in AI-resistant breast cancer cells. (**A**) X-ray crystal structures of STAT1 and ERα were downloaded from PDB and prepared for docking by removing ligands, water molecules and extra chain of amino acids. Chain A was selected for both proteins. The proteins were further prepared using MGL tools. The final prepared protein structures were uploaded to the GRAMM-X protein docking server for checking interactions. The final output file was analyzed using PYMOL program. (**B**) T47D, MCF-7, and MCF-7:5C cells were immunoprecipitated with anti-ERα or rabbit IgG and immunoblotted for STAT1, STAT2, IRF9 and ERα. (**C**) All cell lines were grown on coverslips and processed with the Duolink^®^ PLA Fluorescence with ERα and STAT1 antibodies.

**Figure 6 cancers-13-05130-f006:**
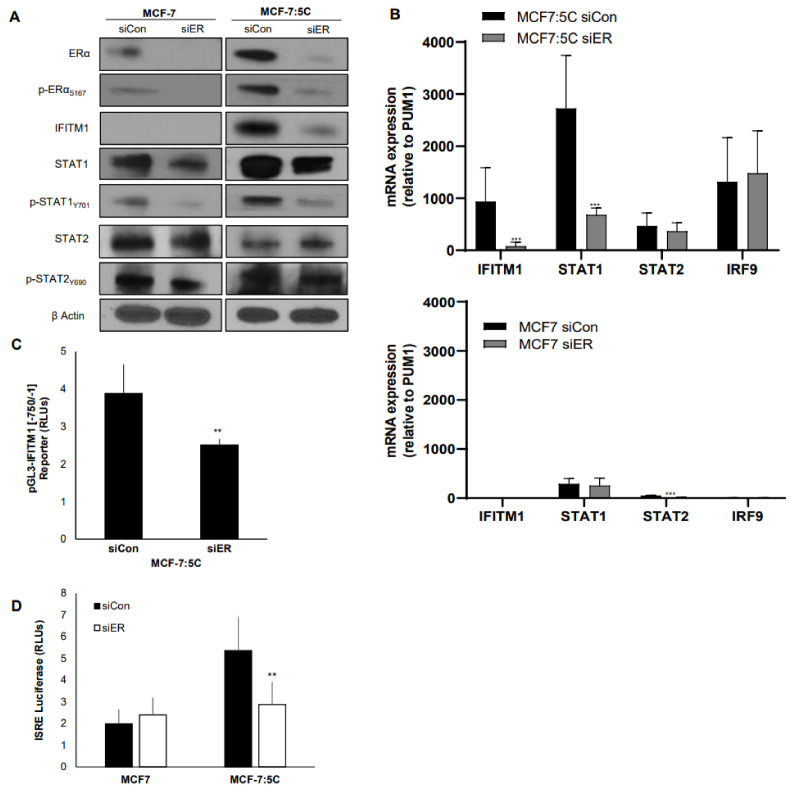
Inhibition of ERα directly affects IFITM1, a downstream target of IFNα signaling. MCF-7 and MCF-7:5C cells were transiently transfected for 48 h with siRNA against ERα and (**A**) immunoblotted for the proteins indicated; (**B**) analyzed by RT-PCR. (**C**) MCF-7:5C cells were transiently transfected for the IFITM1 promoter construct and siRNA against ERα for 48 h. (**D**) MCF-7 and MCF-7:5C cells were transfected with the ISRE reporter construct and siRNA against ERα for 48 h. Luciferase activity was read. ** *p* < 0.05 *** *p* < 0.001.

**Figure 7 cancers-13-05130-f007:**
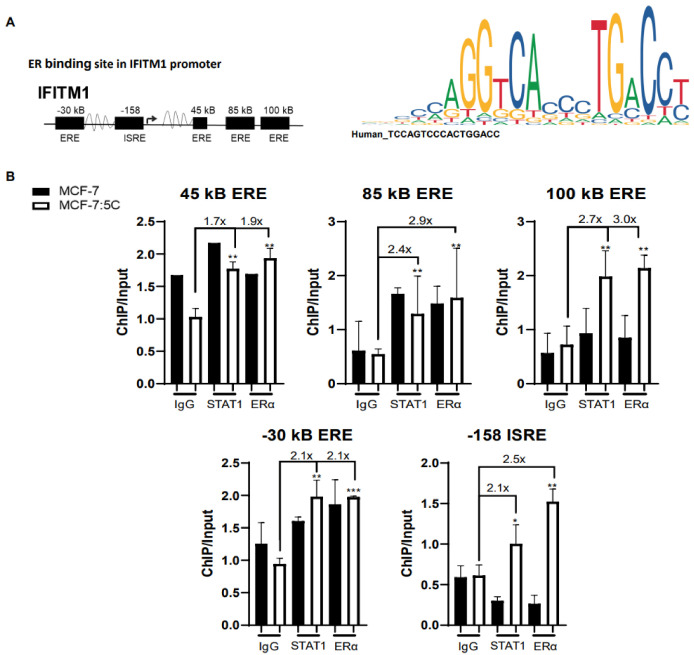
ERα and STAT1 regulate IFITM1 through binding to ERE and ISRE elements in the promoter. (**A**) ChIP data from Tam-resistant breast cancer cells from the UCSC Genome Browser were analyzed for potential ERα binding sites. (**B**) Chromatin immunoprecipitation (ChIP) with antibodies against ERα, STAT1 or species-specific IgG control was performed and analyzed by qPCR and DNA gels on the isolated DNA using primers designed to amplify the ERE and ISRE regulatory regions. Recruitment of the indicated proteins to the ERE and ISRE site was compared to input DNA and displayed as mean ± SD of technical triplicates in two independent experiments. * *p* < 0.5, ** *p* < 0.05 and *** *p* < 0.001.

**Figure 8 cancers-13-05130-f008:**
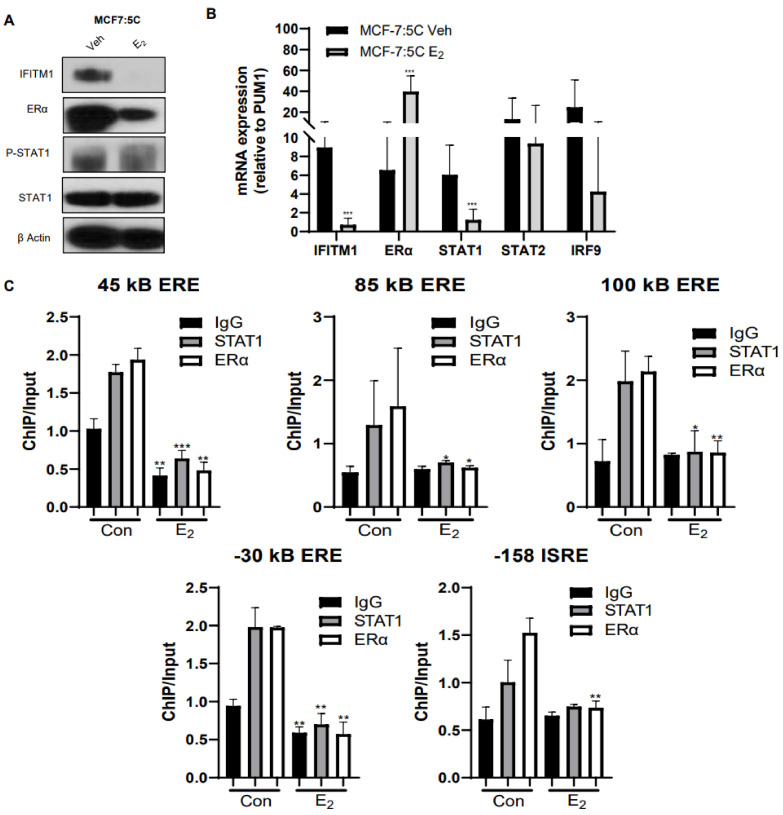
E_2_ treatment inhibits IFITM1 expression and blocks ERα and STAT1 recruitment to the IFITM1 promoter. MCF-7:5C cells were treated for 48 h with E_2_ and (**A**) immunoblotted for ERα, p-STAT1, STAT1, and IFITM1 expression; (**B**) analyzed by RT-PCR. (**C**) Chromatin immunoprecipitation (ChIP) with antibodies against ERα, STAT1 or species-specific IgG control was performed and analyzed by qPCR and DNA gels on the isolated DNA using primers designed to amplify the ERE and ISRE regulatory regions. Recruitment of the indicated proteins to the ERE and ISRE site was compared to input DNA and displayed as mean ± SD of technical triplicates in two independent experiments. * *p* < 0.5, ** *p* < 0.05 and *** *p* < 0.001.

**Figure 9 cancers-13-05130-f009:**
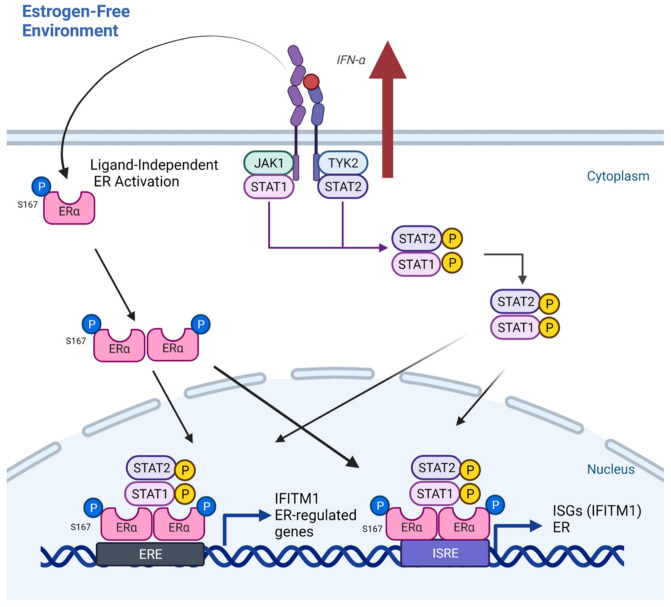
Proposed mechanism of enhanced IFNα signaling on ligand-independent expression of ERα in driving AI resistance and IFITM1 expression. Enhanced IFNα signaling seen in the AI-resistant MCF-7:5C cells upregulates JAK/STAT signaling and expression of not only ISGs but also ERα. This enhanced signaling also promotes ligand-independent activation of ERα through phosphorylation of the S167 residue. STAT1 and ERα then function as co-activators of not only ER-regulated genes but also of IFITM1 by binding directly to its promoter which increases survival signaling in AI-resistant cells. (Figure created with Biorender.com.)

## Data Availability

The data presented in this study are available from the corresponding author upon request.

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
