# Peer review of "Enhanced IFNα Signaling Promotes Ligand-Independent Activation of ERα to Promote Aromatase Inhibitor Resistance in Breast Cancer"

_cancers, 2021, doi:10.3390/cancers13205130_

Round 1
Reviewer 1 Report
Introduction is slighty brief but concise and provides a profitable backround including relevant references. Experimental design is appropiate, and the methods are enough and adequately described and explained. Some figures are too close to each other, but this does not make them difficult to understand. Conclusions include a very profitable figure which show the different pathways analyzed in the study.
Minor comments:
- First line of Reference 1 is not lined up with the others.
- A dot at the end of the paragraph Acknowledgments is missed.
- If you use A, B, C... in the figures, you should use A, B, C... in the text below the figure, instead of a, b,c...
- Some pictures of western blots seem like any part of the membrane (figures 2 or 6 for example) Can we see the whole picture of these membranes?
Author Response
Reviewer #1 (Reviewer Comments to Author):
Introduction is slightly brief but concise and provides a profitable background including relevant references. Experimental design is appropriate, and the methods are enough and adequately described and explained. Some figures are too close to each other, but this does not make them difficult to understand. Conclusions include a very profitable figure which show the different pathways analyzed in the study.
Minor comments:
- First line of Reference 1 is not lined up with the others.
- A dot at the end of the paragraph Acknowledgments is missed.
- If you use A, B, C... in the figures, you should use A, B, C... in the text below the figure, instead of a, b,c...
- Some pictures of western blots seem like any part of the membrane (figures 2 or 6 for example) Can we see the whole picture of these membranes?
Response #1: Thank you, we have made the changes to the formatting errors indicated, for the first three comments. For the final comment, we uploaded the whole Western film which will be included in the supplemental data as requested for the journal requirements. The Western blot films were uploaded as a zipped file and is available along with the supplemental data.
Reviewer 2 Report
Overall, this study is very organised and the manuscript is well written. The methodology is clear and results are well structured.
Discussion:
- This study is a follow-up investigation to already established findings, the novelty of this study lies in elucidating the regulation of ERα and STAT1 in AI resistance in breast cancer cell lines. It would be helpful for the readers to understand the significance of this study if the authors could expand the discussion section to explain the potential of this study. How can the findings from this study be translated to a clinical setting? How do you see this area unfolding in the forthcoming years? It would be extremely interesting for the readers, especially considering the changing landscape of medical treatment in post-menopausal women with ER+ breast cancer.
- What are the limitations of this study?
Author Response
Reviewer #2 (Reviewer Comments to Author):
Overall, this study is very organized, and the manuscript is well written. The methodology is clear, and results are well structured.
Discussion:
- This study is a follow-up investigation to already established findings, the novelty of this study lies in elucidating the regulation of ERα and STAT1 in AI resistance in breast cancer cell lines. It would be helpful for the readers to understand the significance of this study if the authors could expand the discussion section to explain the potential of this study. How can the findings from this study be translated to a clinical setting? How do you see this area unfolding in the forthcoming years? It would be extremely interesting for the readers, especially considering the changing landscape of medical treatment in post-menopausal women with ER+ breast cancer.
- What are the limitations of this study?
Response #2: Thank you for your comments. We have included a paragraph that more thoroughly addresses both comments in lines 581-602 of the discussion.
Reviewer 3 Report
The present manuscript adds important information to our understanding of the basic mechanisms involved in resistance to aromatase inhibitors (AIs). Taking the central role of AIs in the treatment algorithms of BC in consideration, the given information is of major importance and may lead to novel drug combinations potentially enhancing the efficacy of AIs. In my point of view, the presentation and discussion of the data is sound and may be published in its current form.
Author Response
Reviewer #3 (Reviewer Comments to Author):
The present manuscript adds important information to our understanding of the basic mechanisms involved in resistance to aromatase inhibitors (AIs). Taking the central role of AIs in the treatment algorithms of BC in consideration, the given information is of major importance and may lead to novel drug combinations potentially enhancing the efficacy of AIs. In my point of view, the presentation and discussion of the data is sound and may be published in its current form.
Response #3: Thank you for your comments.